# Mice with 16p11.2 Deletion and Duplication Show Alterations in Biological Processes Associated with White Matter

**DOI:** 10.3390/ijms26020573

**Published:** 2025-01-11

**Authors:** Tianqi Wang, Megan Sharp, Ilaria Morella, Francesco Bedogni, Vladimir Trajkovski, Riccardo Brambilla, Yasir Ahmed Syed

**Affiliations:** 1Neuroscience and Mental Health Innovation Institute, Cardiff University, Hadyn Ellis Building, Cardiff CF24 4HQ, UK; wangt55@cardiff.ac.uk (T.W.); morellai@cardiff.ac.uk (I.M.); bedognif@cardiff.ac.uk (F.B.); 2School of Bioscience, Cardiff University, The Sir Martin Evans Building, Museum Ave., Cardiff CF10 3AX, UK; 3Department of Biology and Biotechnology “Lazzaro Spallanzani”, University of Pavia, Via Adolfo Ferrata, 9, 27100 Pavia, PV, Italy; 4School of Medicine, Cardiff University, McKenzie House, 30–36 Newport Road, Cardiff CF24 0DE, UK; 5Institute of Special Education and Rehabilitation, Faculty of Philosophy, University “Ss. Cyril and Methodius”, Blvd. Goce Delchev 9A, 1000 Skopje, North Macedonia; vladotra@fzf.ukim.edu.mk

**Keywords:** 16p11.2 deletion, duplication, white matter, copy number variation, autism spectrum disorder, neurodevelopmental disorder

## Abstract

Deletion and duplication in the human 16p11.2 chromosomal region are closely linked to neurodevelopmental disorders, specifically autism spectrum disorder. Data from neuroimaging studies suggest white matter microstructure aberrations across these conditions. In 16p11.2 deletion and duplication carriers, potential gene dosage effects may impact white matter organisation, contributing to phenotypes including impaired cognition. However, the biological mechanisms underlying this white matter pathology remain unclear. To bridge this knowledge gap, we utilised mouse models of 16p11.2 deletion and duplication to explore changes in corpus callosum oligodendrocytes, myelination, axon caliber, and astrocytes. Immunofluorescence staining was employed to measure lineage and mature oligodendrocyte numbers, as well as myelin basic protein and glial fibrillary acidic protein fluorescence intensity. Transmission electron microscopy was utilised to evaluate axonal structural alterations related to myelin, such as myelinated axon percentage, diameter, myelin thickness, and g-ratio. Our findings reveal changes in the number of mature oligodendrocytes, myelination levels, axon diameter, and astrocytes in the corpus callosum of mice with 16p11.2 deletion and duplication. Deletion mice displayed a tendency toward reduced counts of mature oligodendrocytes and myelination levels, while duplication mice exhibited a notable increase. Axon diameter variations included a significant increase in axon diameter and myelin thickness in both deletion and duplication mice, but with irregular structure in duplication mice. Variances in astrocytes between genotypes showed significant early increases in development for both deletion and duplication mice compared to wild-type mice, with this rise sustained in duplication mice but significantly diminished in deletion mice at a later stage. Our research reveals changes in the biological mechanisms impacting white matter. Comparison of reciprocal trends in 16p11.2 deletion and duplication mice with wild-type mice suggests the possibility of gene dosage effects. Identification of these mechanisms offers an initial step in unveiling therapeutic targets for associated neurodevelopmental disorder phenotypes.

## 1. Introduction

Autism spectrum disorder (ASD) is a neurodevelopmental disorder (NDD) characterised by repetitive behaviours and impaired social communication [1]. It represents a significant mental health and public health concern in contemporary society. Approximately 1/100 children globally have ASD, and the lifetime cost of one affected individual is £2.4 million, meaning that ASD is not only debilitating for those with the disorder but places substantial socio-economic burdens on caregivers and agencies [2,3]. Copy number variations (CNVs) have emerged as highly penetrant risk factors predisposing individuals to a variety of NDDs, including ASD [4]. One identified human locus susceptible to CNVs and strongly associated with a variety of NDDs is the 16p11.2 locus, a region of 593 kb containing 27–29 genes [5,6,7,8]. Deletion and duplication at this locus are seen at remarkably higher prevalence in those with NDDs and are one of the most common genetic aetiologies of ASD, accounting for ~1% of cases [5,9,10]. Alongside this, these CNVs confer increased risk of intellectual disability, motor, language, and developmental delays, epilepsy, and attention deficit hyperactivity disorder (ADHD) [11,12]. Interestingly, 16p11.2 deletion is associated with increased symptom severity, macrocephaly, and higher risk of ASD, while duplications are more strongly associated with microcephaly and schizophrenia [13,14]. These phenotypes are thought to manifest because of altered neural and synaptic developmental trajectories, arising due to the loss or duplication of critical genes in this region. This suggests a potential for gene dosage effects on cellular mechanisms, mediating the risk of NDDs and their associated phenotypes at the 16p11.2 locus. Understanding the relationship between genotype, presentation of clinical phenotypes, and potential underlying aberrant neurobiological mechanisms is important to better appreciate the potential causes of the clinical phenotypes associated with 16p11.2 CNVs, enabling early intervention for their treatment and management.

Aberrant white matter microstructure has been reported across NDDs, with implications for brain connectivity and cognitive function. In ASD, delayed development of major white matter tracts, including the corpus callosum, has been linked with increased symptom severity, which is also associated with both hypo- and hypermyelination [15,16]. Diffusion tensor imaging (DTI) studies have demonstrated decreased fractional anisotropy (FA), a measure of integrity and organisation of white matter tracts whereby decreased FA implies disorganisation, in individuals with ASD, ADHD, schizophrenia, and motor disorders [17,18]. Interestingly, DTI studies of children with 16p11.2 deletion have revealed increased FA in major white matter tracts including the corpus callosum, with this suggested to underlie the phenotype of impaired cognition [19]. A further study encompassing both 16p11.2 deletion and duplication carriers revealed reciprocal changes in white matter organisation, with deletion carriers exhibiting increased FA and duplication carriers exhibiting decreased FA [20]. These imaging studies highlight superficial differences in white matter microstructure across 16p11.2 deletion and duplication carriers, suggesting that alterations at this locus impact white matter development. However, they do not elucidate the biological mechanisms behind this aberrant development, leaving gaps in our understanding of how it arises and relates to clinical phenotypes.

White matter is composed of bundles of axons ensheathed in myelin, essential for high-speed transmission of electrical impulses [21]. Myelin is produced by mature oligodendrocytes through a complex process that begins with the specification of neural progenitor cells into oligodendrocyte progenitor cells. These progenitor cells then proliferate, migrate, and differentiate into mature oligodendrocytes, which ultimately produce myelin. Maintenance and effective functioning of oligodendrocytes is aided by astrocytes, providing energy in the form of brain-derived lactate and regulating glutamate concentrations [22]. Disruption to these processes may result in alterations to myelination and subsequent neuronal function and connectivity, increasing the risk of NDDs [23,24]. Critically, MAPK/ERK signalling has been linked to these processes of oligodendrocyte differentiation, survival, and myelin synthesis [23]. Three genes located at the 16p11.2 locus—*MAPK3, TAOK2*, and MVP—are posited to be implicated in this signalling pathway [8,25,26,27]. Reduced Erk1/2-MAPK signalling has been linked to hypomyelination, suggesting the possibility of white matter-specific gene dosage effects in 16p11.2 deletion and duplication carriers, potentially influencing connectivity and clinical phenotypes.

Interestingly, a mouse model of 16p11.2 deletion exploring functional connectivity revealed increased axon diameters, myelin thickness, and g-ratios in corpus callosum axons [28]. Despite this, another 16p11.2 deletion mouse model revealed decreased expression of myelin-related genes in the striatum and reduced myelin thickness [29]. Additionally, a rat model of ASD exploring the biological mechanisms underlying white matter pathology demonstrated reduced myelin thickness and reductions in lineage and mature oligodendrocytes related to behavioural changes [30]. These findings highlight the potential for altered biological mechanisms related to white matter development, suggesting that the 16p11.2 locus plays a critical role in this, with deviations from normal dosage resulting in pathology potentially related to associated NDDs, including ASD.

In this study, we aimed to elucidate the biological mechanisms underlying the white matter pathology seen in 16p11.2 deletion and duplication carriers. Given the reciprocal changes observed in DTI studies and the potential for gene dosage effects, we hypothesised that there would be distinct reciprocal alterations in numbers of oligodendrocytes, myelination, axonal architecture, and astrocytes between 16p11.2 WT, deletion, and duplication mice. We utilised mouse models of these CNVs to explore the impact of age on oligodendrocyte development, myelination, myelin-related axonal architecture, and astrocytes in the corpus callosum, the largest white matter tract in the brain. Our analysis found consistent decreases in the number of mature oligodendrocytes and myelination in the 16p11.2 deletion mice. By contrast, significant increases in the number of mature oligodendrocytes and myelination were observed in the duplication mice, along with altered axon calibers. These findings suggest abnormal biological mechanisms that could contribute to neurodevelopmental disorder (NDD) phenotypes associated with 16p11.2 copy number variations.

## 2. Results

We utilised mice models of 16p11.2 deletion and duplication to assess changes in the biological mechanisms related to white matter in the corpus callosum (Figure 1). Mice carrying heterozygous deletion or duplication of the 16p11.2 region were crossed with wild-type C57BL/6N mice to obtain experimental heterozygotes and wild-type littermates.

### 2.1. 16p11.2 Duplication Mice Have More Mature Oligodendrocytes in the Corpus Callosum

As myelin is produced by mature oligodendrocytes and certain genes at the 16p11.2 locus may be involved in oligodendrocyte formation, we first investigated whether genotype influenced the number and/or maturation of oligodendrocytes in the corpus callosum. Coronal sections of the corpus callosum from each mouse were stained for OLIG2 and CC1 proteins. Cells stained for OLIG2 alone marked all of the oligodendrocyte lineage cells, from early progenitors to mature cells. Co-stained cells for OLIG2/CC1 represented specifically mature, myelin-producing oligodendrocytes (Figure 2).

Quantification of this immunofluorescence staining revealed no significant differences in the numbers of oligodendrocyte lineage cells between the 16p11.2 deletion or duplication mice compared with the WT mice at either the P21 or P90 time points (Figure 2). Despite this, we observed differences in the number of mature oligodendrocytes between the genotypes. Notably, we found a significant increase (*p* < 0.01) in numbers of mature oligodendrocytes in the 16p11.2 duplication mice at P21 compared to the WT mice. Whilst we found no significant differences between the 16p11.2 deletion and WT mice at P21, the deletion mice presented a slightly elevated number of mature oligodendrocytes compared to the WT mice (Figure 2). We also identified no significant differences between the number of mature oligodendrocytes at P90 when comparing either the 16p11.2 deletion or duplication mice with the WT mice (Figure 2). Despite being non-significant, slight alterations in the numbers of mature oligodendrocytes between the genotypes at P90 can be seen visually in Figure 2. Here, the 16p11.2 duplication mice have marginally higher numbers of mature oligodendrocytes compared to the WT mice, whereas the deletion mice have fewer numbers. These alterations in the number of mature oligodendrocyte cells may in turn have a noticeable influence on the extent of myelination in the corpus callosum, especially at P21 in the duplication mice where statistical significance was observed, assessed using LME models. Overall, the numbers of both lineage and mature oligodendrocytes appeared to increase across the genotypes from P21 to P90.

### 2.2. 16p11.2 Duplication Mice Have Increased Myelination in the Corpus Callosum

Given the differences in numbers of mature (myelin-producing) oligodendrocytes, we next examined the effect of genotype on the extent of myelination in the corpus callosum at P21 and P90. We stained coronal sections from each mouse for MBP, a protein localised to myelin sheaths, and quantified its fluorescence intensity. We found a significant increase in MBP fluorescence intensity in the 16p11.2 duplication mice compared with the WT mice at both P21 (*p* < 0.01) and P90 (*p* < 0.05) (Figure 3). This suggested a significant increase in the extent of corpus callosum myelination in the 16p11.2 duplication mice. The representative images in Figure 3 distinctly display this difference in MBP fluorescence. No significant differences were identified between the 16p11.2 WT and deletion mice at either time point, although a slight decrease in MBP fluorescence intensity was apparent at P90 (Figure 3). Overall, the levels of MBP fluorescence intensity presented across the P21 and P90 time points were relatively consistent, implying little change in the extent of myelination between these developmental stages.

### 2.3. 16p11.2 CNV Mice Have Altered Callosal Axon Calibre

We then investigated whether any structural changes at the level of the axon occurred between the genotypes. To assess structural alterations related to myelin, we performed an ultra-structural analysis of axons in the corpus callosum using TEM. We calculated the percentage of myelinated axons in each image and measured the axon diameter, myelin thickness, and g-ratio of each myelinated axon, using ordinary one-way ANOVA to analyse the effect of genotype on these measures (Figure 4).

The analyses revealed a significant decrease in the percentage of myelinated axons in the corpus callosum of the 16p11.2 deletion mice when compared to the WT mice (Figure 4c, *p* < 0.01). Conversely, we found a significant increase in the percentage of myelinated axons in the corpus callosum of the 16p11.2 duplication mice when compared to the WT mice (Figure 4c, *p* < 0.01). These results illustrated mirrored percentages of the myelinated axons in the corpus callosum of the 16p11.2 deletion and duplication mice compared to the WT mice, highlighting a difference in the constitution of the corpus callosum regarding myelinated axon coverage.

Additionally, we observed significant differences in axon calibre between the WT and deletion and duplication genotypes. Both the 16p11.2 deletion (*p* < 0.001) and duplication (*p* < 0.001) mice exhibited significant increases in axon diameter compared to the WT mice (Figure 4d). Despite increased axon diameters in the 16p11.2 deletion mice, the myelinated fibres appeared to maintain a relatively regular structure (Figure 4a). The duplication mice also exhibited increased axon diameters compared to the WT mice (Figure 4d); however, we immediately noticed that many of the myelinated fibres appeared elongated with irregular morphology (Figure 4a). This demonstrated differences in axon diameter between the genotypes that may have the capacity to influence myelination. We further explored the relationship via the g-ratio, examining the association between naked axon diameter and myelinated axon diameter (Figure 4b). We found a significant decrease in g-ratio for the 16p11.2 deletion (*p* < 0.001) and duplication (*p* < 0.001) mice compared with the WT mice (Figure 4e), potentially influenced by their distinct axon calibre. However, considerable individual variation in axon diameter and myelin thickness led to genotype overlap. Figure 4f aids the visualisation of the relationship between axon diameter and myelin thickness, depicting the distribution of each measured axon across the genotypes. Figure 4f reveals smaller g-ratios in the 16p11.2 deletion and duplication mice compared to the WT mice, with Figure 4g providing visualisation of this across all axons. Smaller g-ratios generally indicate greater myelin thickness and can also be an indicator of axonal integrity and function.

### 2.4. 16p11.2 CNV Mice Have Altered Numbers of Astrocytes

Given the importance of astrocytes in supporting oligodendrocytes, we performed further immunofluorescence staining for GFAP to investigate the effect of genotype on astrocyte populations. We found a significant increase in GFAP fluorescence intensity for both the 16p11.2 deletion (*p* < 0.001) and duplication (*p* < 0.01) mice compared to the WT mice at P21 (Figure 5). This suggested a significant increase in astrocytes in both the deletion and duplication mice at the P21 time point. At P90, significant differences were also identified. Here, however, we found a significant decrease in GFAP fluorescence intensity for the 16p11.2 deletion mice (*p* < 0.05) and a significant increase for the duplication mice (*p* < 0.001) (Figure 5). This suggested a significant reduction in astrocytes for the deletion mice compared to the WT mice at P90, while the duplication mice continued to exhibit increased numbers of astrocytes. Additionally, we observed an apparent decrease in GFAP fluorescence intensity across the genotypes between the P21 and P90 time points. This implied there were fewer astrocytes in all genotypes at this later time point.

## 3. Discussion

In this study, we utilised mouse models with 16p11.2 deletion and duplication to investigate changes in the biological mechanisms affecting white matter abnormalities found in individuals with these copy number variations (CNVs). Our findings clearly reveal modifications in white matter mechanisms, such as the number of mature oligodendrocytes, myelination levels, axon size, and astrocytes in the corpus callosum of mice with 16p11.2 deletion and duplication. Comparing these measures, we observed contrasting trends between the 16p11.2 deletion and duplication mice versus the wild-type mice. These results shed light on the potential impact of gene dosage on the biological mechanisms that influence white matter, possibly contributing to its pathology and related characteristics in individuals with 16p11.2 CNVs.

A key observation from the immunofluorescence staining of oligodendrocytes revealed no significant variances at P21 or P90 in the quantities of oligodendrocyte lineage cells in the corpus callosum of the 16p11.2 deletion or duplication mice compared to the wild-type mice. These results align with the study by Ju et al., which also found no notable distinctions in the number of oligodendrocyte lineage cells in 16p11.2 deletion mice, specifically in the striatum at P60 [29]. These findings indicated consistent specification, growth, and movement of oligodendrocyte progenitor cells in both the striatum and corpus callosum of the 16p11.2 deletion mice, as well as in the corpus callosum of the duplication mice. The absence of abnormal changes in the quantity of oligodendrocyte lineage cells across the genotypes implies that this is not a mechanism contributing to atypical white matter conditions in 16p11.2 CNV carriers.

In our study, we explored the oligodendrocyte populations further and identified a significant increase in mature oligodendrocytes in the 16p11.2 duplication mice at P21 compared to the wild-type mice. While the trends were not statistically significant, they persisted to some extent at P90. On the other hand, the deletion mice did not show significant differences; however, a slight decrease in numbers was observed at P90, suggesting potential reciprocal effects of genotype on oligodendrocyte maturity. These results suggest that disturbances at the 16p11.2 locus impact oligodendrocyte differentiation, revealing a mechanism that could influence and contribute to white matter disorders. Additionally, they demonstrate that age does not play a role in overcoming the initial aberrant differentiation we observed. Oligodendrocyte differentiation is a multifaceted process with crucial 16p11.2 genes, such as *MAPK3, MVP*, and *TAOK2*, believed to affect the MAPK/ERK signaling pathways that play a role in regulating oligodendrocyte differentiation [23,31]. The respective loss or duplication of genes at this locus may therefore have reciprocal reduction and acceleration effects on the differentiation of oligodendrocytes. This is plausible, as increased activation of MAPK signalling has been associated with enhanced oligodendrocyte progenitor cell differentiation [32]. In vivo experiments have shown that reduced Erk1/2-MAPK activity is associated with hypomyelination, a condition linked to ASD traits like impaired social behaviour [33,34]. Considering the potential gene dosage effects, it is conceivable that deletion in the 16p11.2 region may lead to reduced myelination, potentially contributing to the strong link of this copy number variation (CNV) with autism spectrum disorder (ASD). Conversely, it could be postulated that duplications in the 16p11.2 region may result in increased myelination, building upon the initial hypothesis of reciprocal changes in mature oligodendrocytes and myelination. Further investigation is needed to ascertain the genetic mechanisms involved in 16p11.2 CNV white matter development; however, our study identifies a biological mechanism with potential to impact white matter development in the corpus callosum and shows that age has little impact on overcoming white matter alterations observed in early development.

Our evaluation of MBP fluorescence intensity, which highlights myelin sheaths, uncovered changes in corpus callosum myelination consistent with trends in mature oligodendrocyte numbers. The mice with 16p11.2 duplication exhibited notably heightened MBP intensity at both P21 and P90, suggesting hypermyelination. Conversely, the deletion mice displayed insignificant reductions at both time points, indicating hypomyelination that was not improved with age. Hypo- and hypermyelination of white matter tracts have been observed in both mouse models and individuals with ASD [15,35]. As ASD is a common life-long phenotype in deletion and duplication carriers, the present study suggests altered underlying mechanisms of myelination that converge onto the same phenotype throughout development. However, macrocephaly, commonly exhibited in deletion carriers, is associated with hypermyelination, whilst microcephaly, manifested in duplication carriers, could be associated with hypomyelination [13,35,36]. The current study challenges the typical trend by revealing hypermyelination in the 16p11.2 duplication mice and hypomyelination in their deletion counterparts. This divergence indicates that the processes linked to 16p11.2-related micro- and macrocephaly, influenced by *KCTD13, may function separately from those governing myelination. This highlights the need for further exploration into the precise roles and interactions of 16p11.2 genes in the development of white matter.

Interestingly, we observed an overall increase in the numbers of lineage and mature oligodendrocytes across the genotypes from P21 to P90. However, no corresponding increase in MBP fluorescence was observed, indicating the limited impact of ongoing oligodendrocyte proliferation and differentiation seen post-P21 on corpus callosum myelination. This suggests that factors beyond mature oligodendrocyte numbers may influence myelination, such as reduced myelin gene/protein expression, as observed in the 16p11.2 deletion mice at P60 [29]. Additionally synaptic communication has been hypothesised to stimulate myelination [21]. With many 16p11.2 genes involved in synapse morphology, and an evident reduction in synaptic density in 16p11.2 deletion and duplication human induced pluripotent stem cells (iPSCs), it is plausible that synaptic disruption influences the capacity for myelination during development [9,37]. This complex relationship between gene expression, synapses, and potential functional connectivity aids the explanation of unchanged myelination despite increased numbers of mature oligodendrocytes. Further investigation of the relationship between these processes may unveil additional mechanisms influencing aberrant white matter development, potentially impacting clinical phenotypes.

The contrasting levels of corpus callosum myelination, reduced in the 16p11.2 deletion mice and elevated in the duplication mice, highlight potential alterations to white matter microstructure. Our ultra-structural analysis of callosal axons revealed significant differences in myelin-related axon structure across these mice. Specifically, significantly fewer myelinated axons were observed in the deletion mice, whilst significantly more were observed in the duplication mice. These findings align with our previous observed trends for mature oligodendrocyte numbers and myelination extent, indicating a consistent decrease in myelin in the deletion mice and increase in the duplication mice. This insight into myelin-related changes at the axon level offers a mechanical explanation for the reciprocal myelination levels seen in the 16p11.2 deletion and duplication mice. Despite the use of a different mouse cohort for TEM analysis, the consistency of the findings with oligodendrocyte and myelination trends suggests robustness in the mouse models employed.

Further analysis of axon structure revealed significantly increased myelin thickness in the 16p11.2 deletion mice, whilst significantly decreased myelin thickness in the duplication mice. Myelin thickness is thought to be influenced by axon diameter, with larger axons generally exhibiting greater myelin thickness [38]. We reported a significant increase in axon diameter for both deletion and duplication genotypes compared to WT; however, callosal axons in the duplication mice appeared to have irregular morphology. This could limit their capacity for effective myelination and possibly reduce their structural integrity and subsequent g-ratio, as demonstrated in our results. Human 16p11.2 deletion and duplication carriers exhibit respective increases and decreases in FA, associated with cognitive and behavioural alterations [20]. Increased FA suggests greater organisation of white matter tracts, with the increased myelin thickness and axon diameter observed in the 16p11.2 deletion mice offering an explanation behind the increased FA in human carriers. These findings align with those of Bertero et al., who performed DTI in 16p11.2 deletion mice, revealing widespread increases in FA along with increased axon diameter and myelin thickness [28]. However, they posited that these changes were too widespread to be a direct correlate of the impaired functional hypoconnectivity hypothesised to underlie the socio-cognitive deficits in deletion carriers observed in their study. The results from our study could attribute this decreased hypoconnectivity to the decreased number of myelinated axons observed in the deletion mice. The decreased myelin thickness and abnormal morphology of myelinated fibres in the duplication mice could underlie the decreased FA seen in human 16p11.2 duplication carriers. Decreased FA has also been consistently reported in schizophrenia, for which the 16p11.2 duplication confers greater risk [17]. This analysis provides a clear mechanism by which alterations in axon structure between the 16p11.2 deletion and duplication mice may underlie superficial measures of altered white matter integrity seen and associated with NDD phenotypes in 16p11.2 CNV carriers.

Although mature oligodendrocytes produce myelin, astrocytes are posited to play a critical role in facilitating and maintaining myelination via their reactive nature and providing support to oligodendrocytes [39,40]. Interestingly, our analysis indicated there were significantly more astrocytes in the deletion and duplication mice compared to the WT mice at P21. However, at P90, we observed a significant decrease in the number of astrocytes in the deletion mice, whilst the increase remained in the duplication mice compared to the WT mice. This appears to follow the trends we identified regarding mature oligodendrocyte numbers, suggesting that there could be an association between astrocyte activity and oligodendrocyte maturity and myelination potential. Despite this, astrocytes are involved in the regulation of many key aspects of brain development, including neuronal migration, dendritic and spine development, and synapse formation [41]. Postmortem studies of those with ASD have revealed abnormalities in astrocytes and research suggests that cognitive deficit and repetitive behaviour can be induced by ASD iPSC astrocytes alone [42]. This suggests that astrocytes may be aberrant in ASD. Given the increased numbers of astrocytes in the 16p11.2 deletion and duplication mice at P21, and that increased risk of ASD is apparent for both CNV carriers, it would be beneficial to explore the impact of astrocyte function in 16p11.2 CNV carriers more widely, as well as in relation to white matter pathology.

While mouse models of 16p11.2 deletion and duplication are commonly utilised to understand aberrant neurodevelopmental mechanisms, it is crucial to acknowledge that the generalisability of the findings may be restricted by species-specific biases [43]. Bertero et al. showcased the robust translational relevance of the 16p11.2 deletion mouse model in relation to white matter microstructure. This validation endorses the model’s utility in investigating the mechanisms that contribute to white matter deficits [28]. Further studies using human models are needed to confirm the translational validity of our findings.

Another potential limitation of our study is the use of only two mice per sex per genotype. Notably, there was significant variability among mice of the same genotype, especially in the cases of the 16p11.2 WT and deletion mice. This variability could account for the lack of statistical significance observed between the WT and deletion mice in terms of oligodendrocyte cell numbers and MBP fluorescence intensity. It is plausible that sex distinctions might impact gene expression, leading to substantial diversity among individuals. Despite attempts to accommodate biological diversity in the statistical model, including sex as a variable in statistical analyses could help mitigate sex-specific influences in future research. Moreover, enlarging the sample size would strengthen the study’s robustness, enabling the evaluation of significant reciprocal effects for deletion mice compared to duplication mice.

## 4. Materials and Methods

### 4.1. Mouse Model

The 16p11.2 deletion and duplication mouse models were generated by Dr. Herault’s group, as shown in [44]. After rederivation at Cardiff University, mice were bred in a pure C57BL/6N background by crossing heterozygotes with commercial C57BL/6N wild-type mice (Charles River, Margate, UK). To perform genotyping, genomic DNA was extracted from ear clips using the Genlute kit for mammalian genomic DNA (Sigma Aldrich, Gillingham, UK). Deletion was identified by PCR using the following primers: forward (5′-CCTGTGTGTATTCTCAGCCTCAGGATG-3′) and reverse (5′-GGACACACAGGAGAGCTATCCAGGTC-3′). PCR was performed using the following program: 95 °C/4 min, 35 × (94 °C/30 s, 62 °C/30 s, 72 °C/1 min), 72 °C/7 min. Duplication was identified using the following primers: forward (5′-TCACCTAACTTCTTCCCTCTTT-3′) and reverse (5′-CTAGAGAATAGGAACTTCGTTTAAAC-3′). PCR was performed using the following program: 95 °C/4 min, 35 × (94 °C/30 s, 60 °C/30 s, 72 °C/1 min), 72 °C/7 min.

Two different time points, post-natal day 21 (P21) and post-natal day 90 (P90), were examined to determine if age played a role in overcoming the initial aberrant differentiation. P21 marks an early critical point in mice development, while P90 offers a later developmental stage. All experimental procedures were performed in accordance with institutional animal welfare, ethical, and ARRIVE guidelines and under the UK Home Office License PPL P0EA855DA (Animals (Scientific Procedures) Act 1986).

### 4.2. Immunofluorescence Staining

Mice (*n* = 24) were used for immunofluorescence staining, with 12 (16p11.2 wild-type (WT) *n* = 4, deletion *n* = 4, duplication *n* = 4) being euthanised at either P21 or P90. The methods for immunofluorescence staining were similar to those previously published [30]. Mice were intracardially perfused with phosphate-buffered saline (PBS) containing 4% paraformaldehyde. After tissue harvest, the brain was post-fixed in paraformaldehyde for 4 h on a shaker and placed in phosphate-buffered 30% sucrose. A cryostat (CM1860 UV, Leica, London, UK) was used to obtain 15 μm thick coronal cryosections of the brain. Coronal slices were mounted onto poly-L-lysine-coated slides (Sigma-Aldrich Company, UK); three sections per mouse per slide, stored at −20 °C until further use.

For staining, the following primary antibodies were used: anti-OLIG2 (AB109186, Abcam, Cambridge, UK) 1:500, anti-APC [CC1] (AB16794, Abcam) 1:500, anti-MBP (Myelin Basic Protein) (MAB386, Merck, Darmstadt, Germany) 1:500, and anti-GFAP (glial fibrillary acidic protein) (Z0334, Dako). Slides for OLIG2 and CC1 co-staining were placed in 5% citrate antigen retrieval buffer (pH 6, 10×, Sigma-Aldrich Company, UK) and heated in a water bath at 90 °C for 15 min. All slides for staining (OLIG2 and CC1 co-staining, MBP and GFAP) were washed for 10 min in 0.1% Triton X-100 (A16046, Alfa Aesar, Heysham, UK) in PBS and blocked for 1 h with 250 μL of blocking and permeabilisation solution: 5% normal goat serum (AB7481, Abcam) in 0.1% Triton X-100 in PBS. The appropriate primary antibody solution was applied, and the slides were incubated overnight at 4 °C. The next day, the slides were washed 3 × 5 min with 0.1% Triton X-100 in PBS. The following appropriate secondary antibodies were added: 555 Anti-Rabbit IgG, 448 Anti-Mouse IgG, 488 Anti-rat IgG, and 488 Anti-rat IgG 1:1000 (Alexa Fluor Life Technologies, Manchester, UK) for OLIG2, CC1, MBP, and GFAP respectively. The slides were incubated for 2 h in darkness at 37 °C, then washed and counter-stained with DAPI for 2 min before being mounted and cover-slipped [45]. To quantify OLIG2+ and CC1+ cells, 5 images from random visual fields of the corpus callosum of each mouse were taken using an inverted fluorescence microscope (Leica DM6000, Leica, UK) at 40×. For MBP and GFAP images, the same microscope was used to take 3 images of visual fields of the corpus callosum per mouse at 10×.

### 4.3. Transmission Electron Microscopy

A new cohort of 12 mice (16p11.2 WT *n* = 4, deletion *n* = 4, duplication *n* = 4), euthanised at P21, was used for TEM. TEM was performed as previously published [30]. In brief, mice were intracardially perfused with PBS containing 2.5% glutaraldehyde and 2.5% paraformaldehyde. After tissue harvest, the brain was post-fixed in perfusion solution overnight on a shaker. The brain was then embedded in resin to enable 50 nm sections to be taken. Sections were stained with 4% uranyl acetate and lead citrate. TEM was performed to obtain images for quantification. Per mouse, images of at least 5 microscope fields across the extent of the anterior-posterior corpus callosum were taken at 7000× to obtain a representative sample.

### 4.4. Statistical Analysis

For quantification of both immunofluorescence staining and TEM images, Fiji (ImageJ) (version 2.14.0/1.54f) was used. For oligodendrocyte immunofluorescence staining, a total of 120 images (P21 *n* = 60: each genotype *n* = 20; P90 *n* = 60: each genotype *n* = 20) were analysed. The total number of OLIG2+ cells (oligodendrocyte lineage cells) and overlapping OLIG2+/CC1+ cells (mature oligodendrocyte cells) were counted when cell bodies were clearly overlapping with DAPI staining. The area of the counted section was taken to calculate the number of lineage (OLIG2+ stained cells) and mature (OLIG2+/CC1+ co-stained cells) oligodendrocytes per mm^2^. MBP and GFAP images were quantified by measuring the mean fluorescence intensity of three random areas of the corpus callosum per image. The DAPI channel was used to identify the corpus callosum area, avoiding bias. For MBP three random areas and for GFAP five random areas from each image were selected to measure the mean intensity per μm^2^. For TEM, a total of 68 images (16p11.2 WT *n* = 22, deletion *n* = 26, duplication *n* = 20) were analysed. The numbers of clearly myelinated and unmyelinated axons per image were counted and the percentage of myelinated axons was calculated. For each myelinated axon, the axon diameter, myelin thickness, and g-ratio were measured. The g-ratio is calculated as $\frac{naked\;axon\;diameter}{myelinated\;axon\;diameter}$, whereby smaller g-ratios indicate greater myelin thickness. A total of 4882 myelinated axons (16p11.2 WT *n* = 1761, deletion *n* = 1492, duplication *n* = 1627) were analysed.

Statistical analysis was completed in GraphPad Prism (version 10.2.3). Ordinary one-way ANOVA analysis was performed to determine differences between the genotypes for the number of OLIG2+ and OLIG2+/CC1+ cells, MBP and GFAP fluorescence intensity, the percentage of myelinated axons, axon diameter, myelin thickness, and g-ratio. These models were considered most appropriate as they enabled the comparison of all data whilst accounting for variations between individual mice. Genotype was considered a fixed factor, as variation was expected. Animal was considered a random factor, as there was no control over the variation among individuals. Within the fixed factor of genotype, levels were created whereby all genotypes were compared to WT, giving two separate results for 16p11.2 WT compared to deletion and duplication. Normality assumptions were assessed visually. Due to skewed data invalidating the normality assumptions for P21 OLIG2+ cells, P90 OLIG2+/CC1+ cells, axon diameter, myelin thickness, g-ratio, and P21 and P90 GFAP fluorescence intensity, these data were log-transformed. All other measures followed a normal distribution. Graphs were created in GraphPad Prism (version 10.2.3).

## 5. Conclusions

Our research highlights that differentiation of oligodendrocytes likely plays a crucial role in the white matter abnormalities seen in individuals with 16p11.2 deletion and duplication. We observed significant differences between the deletion and duplication genotypes compared to the wild type, potentially contributing to white matter issues such as myelination levels, axon size, and astrocyte changes, which persisted with age. By comparing the changes in oligodendrocyte numbers, myelination levels, axon size, and astrocytes between the genotypes, we suggest that there may be gene dosage effects on the corpus callosum. These discoveries offer insights into the white matter problems linked to 16p11.2 copy number variations (CNVs), laying the groundwork for identifying therapeutic targets for related neurodevelopmental conditions like autism spectrum disorder (ASD). Exploring the molecular factors involved in abnormal myelination in the ASD brain could be a key starting point for developing treatments that enhance social skills and address the disruptive behaviours observed in affected individuals.

## Figures and Tables

**Figure 1 ijms-26-00573-f001:**
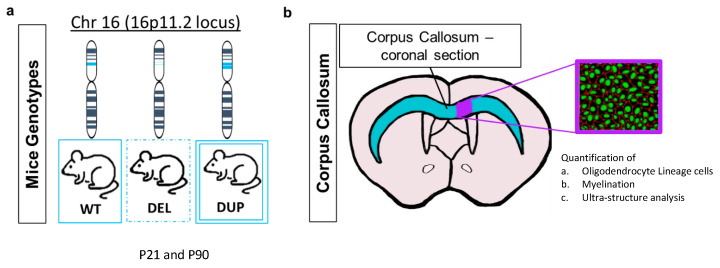
Schematic of mouse genotypes and corpus callosum. (**a**): Mouse genotypes used for each experimental measure. (**b**): Schematic of the corpus callosum, from which the images for analysis were taken. The analysis includes quantification of a. Oligodendrocyte linage cells, b. Myelination, c Ultra-structure analysis of myelin.

**Figure 2 ijms-26-00573-f002:**
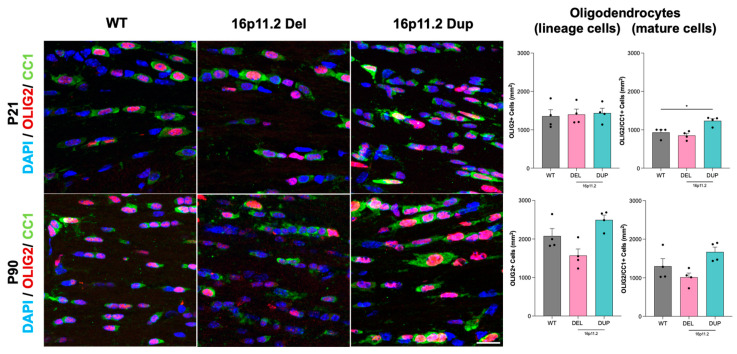
Quantification of oligodendrocytes in the corpus callosum of 16p11.2 WT, deletion, and duplication mice at P21 and P90. Representative images at 40× magnification show immunomarkers DAPI (blue), OLIG2 (red), and CC1 (green) in the corpus callosum of 16p11.2 WT, deletion, and duplication mice at P21 and P90. The dashed square areas are magnified in the top right corner. The average number of oligodendrocyte lineage cells, identified by OLIG2 staining (*n* = 4 mice per genotype per time point) per square millimeter, revealed no significant differences between WT mice and 16p11.2 deletion or duplication mice at P21 or P90. For mature oligodendrocytes, co-stained with OLIG2 and CC1 (*n* = 4 mice per genotype per time point) per square millimeter, there was a notable increase between WT and 16p11.2 duplication mice at P21 (*p* < 0.05 *). However, no significant differences were observed between WT and 16p11.2 duplication mice at P90 or between WT and 16p11.2 deletion mice at either time point. A linear mixed effects model was employed to evaluate differences in the numbers of oligodendrocyte lineage and mature cells while accounting for individual variations between datasets. Data were analysed using one-way ANOVA followed by multiple comparisons with the control group (WT) in Prism (Version 10.2.2). The experiments were conducted independently (*n* = 4), with each experiment involving three mice: 16p11.2 WT, deletion, and 16p11.2 duplication. Results are expressed as the mean ± SEM, with each data point representing an individual mouse. Scale bar: 20 µm.

**Figure 3 ijms-26-00573-f003:**
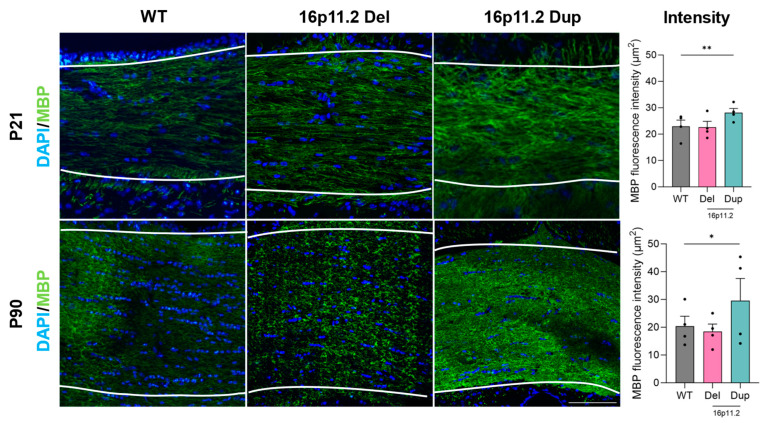
Myelin basic protein fluorescence intensity in the corpus callosum of 16p11.2 WT, deletion, and duplication mice at P21 and P90. Representative images at 10× for immunomarkers DAPI (blue) and MBP (green) in the corpus callosum (marked between the dashed lines) of 16p11.2 WT, deletion, and duplication mice at P21 and P90. Mean MBP fluorescence intensity (μm^2^) (*n* = 4 per genotype per time point). Significant increases in MBP fluorescence are seen between WT and 16p11.2 duplication mice (*p* < 0.01 **) at P21 and P90 (*p* < 0.05 *). Differences between 16p11.2 deletion and duplication mice with WT mice were assessed using a linear mixed effects model. A one-way ANOVA was performed, followed by multiple comparisons against the control group (WT) using Prism (Version 10.2.2). The analysis included data from four independent experiments, each consisting of three mice: 16p11.2 WT, deletion, and 16p11.2 duplication. Results are expressed as the mean ± SEM, with each data point representing an individual mouse. Scale bar: 100 µm.

**Figure 4 ijms-26-00573-f004:**
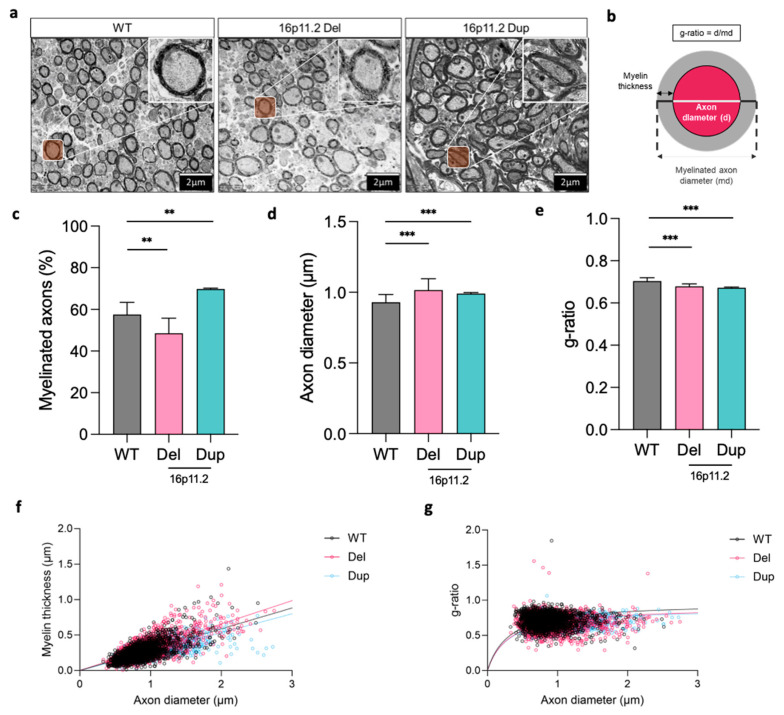
Alterations in myelin specific axon measures across axons in the corpus callosum of 16p11.2 WT, deletion, and duplication mice at P21. (**a**) Representative electron micrographs of axons in the corpus callosum of P21 16p11.2 WT (*n* = 4 mice, *n* = 1761 axons), deletion (*n* = 4 mice, *n* = 1492 axons), and duplication (*n* = 4 mice, *n* = 1627 axons) mice. (**b**) Schematic of measurements taken. (**c**) Mean percentage of myelinated axons per genotype, significant decrease between WT and Del (*p* < 0.01 **) and increase between WT and Dup (*p* < 0.01 **). (**d**) Mean axon diameter of myelinated fibres per genotype, significant increase between WT and Del (*p* < 0.001 ***) and WT and Dup (*p* < 0.001 ***). (**e**) Mean g-ratio per genotype, significant decrease between WT and Del (*p* < 0.001 ***) and WT and Dup (*p* < 0.001 ***). (**f**) Scatter plot of myelin thickness across axon diameter for all myelinated axons per genotype (WT *n* = 1761, Del *n* = 1492, Dup *n* = 1627). (**g**) Scatter plot of g-ratio against axon diameter for all myelinated axons per genotype (WT *n* = 1776, Del *n* = 1492, Dup *n* = 1627). All data were analysed using ordinary one-way ANOVA followed by multiple comparison with the control group (WT) in Prism (Version 10.2.2). Each experiment included four mice across the experimental groups (16p11.2 WT, deletion, and duplication). The data presented are as the mean ± SEM, with each data point representing an individual mouse. Scale bar = 2 µm.

**Figure 5 ijms-26-00573-f005:**
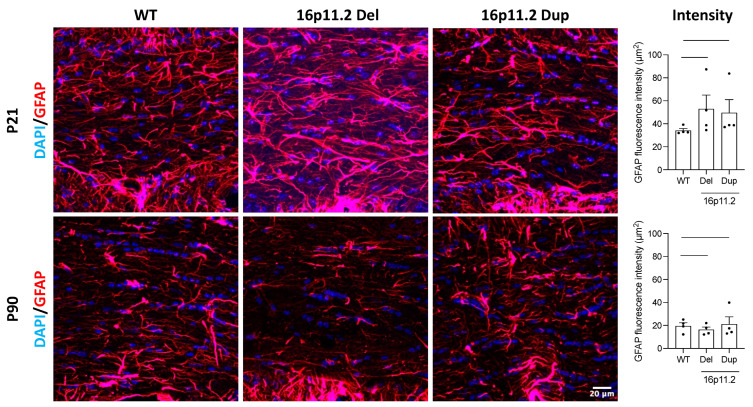
GFAP fluorescence intensity in the corpus callosum of 16p11.2 WT, deletion, and duplication mice at P21 and P90. Representative images at 10× for immunomarkers DAPI (blue) and GFAP (red) in the corpus callosum (marked between the dashed lines) of 16p11.2 WT, deletion, and duplication mice at P21 and P90. Mean GFAP fluorescence intensity (µm^2^) (*n* = 4 per genotype per time point). At P21, significant increases in GFAP are seen between 16p11.2 WT and deletion mice (*p* < 0.001) and duplication mice (*p* < 0.01). At P90, a significant decrease in GFAP is seen between 16p11.2 WT and deletion mice (*p* < 0.05), while a significant increase is seen between 16p11.2 WT and duplication mice (*p* < 0.001). A one-way ANOVA was performed, followed by multiple comparisons against the control group (WT) using Prism (Version 10.2.2). The analysis included data from four independent experiments, each consisting of three mice across the experimental groups: 16p11.2 WT, deletion, and duplication. Results are expressed as the mean ± SEM, with each data point representing an individual mouse. Scale bar: 20 µm.

## Data Availability

The complete dataset is presented in its entirety within the results section of this paper.

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
