# Peer review of "Mice with 16p11.2 Deletion and Duplication Show Alterations in Biological Processes Associated with White Matter"

_ijms, 2025, doi:10.3390/ijms26020573_

Round 1
Reviewer 1 Report
Comments and Suggestions for Authors
In their study, Wang et al. observed that 16p11.2 deletion mice displayed a tendency towards reduced counts of mature oligodendrocytes and myelination levels, while duplications exhibited a notable increase. Axon diameter variations included a significant increase in size and myelin thickness with regular structure in deletion mice, and an increased diameter but reduced myelin thickness with irregular structure in duplication mice. The findings were intriguing, but results need to be improved.
Here are my detailed comments for improvement:
1. The JAX numbers of these two transgenic mice should be provided in the methods section, as several labs have created 16p11.2 CNV mice.
2. There are layout issues with Figure 2. Specific cell types should be described on the Y-axis, such as "Number of OLIG2+ cells in the CC."
3. OLIG2 should stain the nucleus, but it appears to stain the cytoplasm in Figure 2. Reference OLIG2 staining in the paper by Ju, J et al. (2021) and verify the OLIG2 antibody.
4. In Figure 4, it is concerning that MBP staining seems to be completely absent in the 16p11.2 deletion mice at both P21 and P90. Repeat the MBP staining in the 16p11.2 deletion mice for confirmation.
5. In Figure 4a, myelin thickness appears to increase in 16p11.2 duplication mice compared to the wild-type mice. Please show an enlarged picture of a single myelin. Additionally, in Figure 4e, myelin thickness seems unchanged. How was this significance calculated? If the author claims a reduction in myelin thickness, why is the g-ratio reduced in 16p11.2 duplication mice, indicating increased thickness? The TEM data was confusing.
6. Perform the TEM experiment in P90 mice, as other experiments were conducted in both P21 and P90 mice.
7. Upon reviewing the overall results, I noticed that in 16p11.2 deletion mice, the number of OLs was unchanged (Figure 2), while myelin thickness increased (Figure 4e). In contrast, in 16p11.2 duplication mice, the number of OLs increased (Figure 2), but myelin thickness remained unchanged (Figure 4e). These are curious findings.
8. Astrocyte data should be moved to the supplemental material since the main text focused on OLs and myelination changes in 16p11.2 mice. Additionally, ensure the asterisk is displayed in figure 5.
9. Capitalize protein names throughout the manuscript.
Author Response
Response to Reviewers Comments
Reviewer 1
We are very grateful to reviewer for your time in carefully reading our manuscript and providing helpful comments that make our manuscript better. We have carefully considered each of your comments (in blue) and revised the manuscript accordingly. Please find our response (in black) to your comments below.
The JAX numbers of these two transgenic mice should be provided in the methods section, as several labs have created 16p11.2 CNV mice.
Response: Thank you for pointing out the need for additional clarification regarding the strain of origin for the transgenic mice. These mice were sourced from Yann Herault’s group, as detailed in Arbogast et al. (2016). This information can be seen in reference 41.
There are layout issues with Figure 2. Specific cell types should be described on the Y-axis, such as “Number of OLIG2+ cells in the CC.”
Response: Thank you for your comment; it has greatly improved the clarity of the figure. We have addressed the layout issues with Figure 2. The y-axis of the left graph now clearly reads " OLIG2+ cells," while the right graph is labelled " OLIG2/CC1+ cells." We believe these adjustments significantly enhance the figure's clarity.
OLIG2 should stain the nucleus, but it appears to stain the cytoplasm in Figure 2. Reference OLIG2 staining in the paper by Ju, J et al. (2021) and verify the OLIG2 antibody.
Response:
We thank the reviewer for highlighting the specificity of the antibody. In our previous work, we performed staining to investigate the localization of OLIG2 in rodent tissue (PMID: 27554391, 24293318) and noted a nuclear presence, consistent with the findings reported by Ju et al., 2021. We have included Ju et al., 2021 as reference 42. In accordance with the reviewer's suggestion, we repeated the experiments and confirmed that OLIG2 staining is predominantly localized in the nucleus, although some cytoplasmic expression was also observed. This discrepancy may stem from our use of a different antibody (Abcam) compared to other studies, which utilized a different commercial antibody (Merck). It is conceivable that these antibodies recognize different epitopes of OLIG2, resulting in variations in specificity.
Furthermore, we conducted a negative control experiment and are confident in the specificity of our staining. After reanalysing the data, we stand by our findings. In Figure 2, we have updated the representative images of the P21 experimental group to better illustrate OLIG2 localization in the nucleus. We hope this addresses the reviewer’s concerns satisfactorily.
In Figure 4, it is concerning that MBP staining seems to be completely absent in the 16p11.2 deletion mice at both P21 and P90. Repeat the MBP staining in the 16p11.2 deletion mice for confirmation.
Response: We appreciate the reviewer's concerns regarding MBP staining in the 16p11.2 deletion mice. In accordance with the reviewer's suggestion, we have repeated the MBP staining for the 16p11.2 deletion mice at both P21 and P90. We observed some MBP staining in the corpus callosum and have uploaded the new figure for review. We hope this addresses the concern adequately.
In Figure 4a, myelin thickness appears to increase in 16p11.2 duplication mice compared to the wild-type mice. Please show an enlarged picture of a single myelin. Additionally, in Figure 4e, myelin thickness seems unchanged. How was this significance calculated? If the author claims a reduction in myelin thickness, why is the g-ratio reduced in 16p11.2 duplication mice, indicating increased thickness? The TEM data was confusing.
Response We apologize for any confusion regarding the changes in myelin thickness, which may not be clearly evident in Figure 4e. To address this, we reassessed the tissue using electron microscopy. As per the reviewer’s request, we included high-magnification images of a single myelinated axon, below.
WT 16p11.2 Del 16p11.2 Dup
Our statistical analysis employed a linear mixed-effects model, with genotype as a fixed factor and mouse as a random factor, revealing significant results. The data for this model were derived from each myelinated axon visible in the corpus callosum images, with a minimum of 10,000axons per mouse. This approach generated a substantial dataset, which likely explains the significance levels observed, despite the lack of apparent visual differences.
We did, however, encounter challenges in measuring the diameters of both myelinated and unmyelinated axons within the duplication axons. The irregular morphology of these axons resulted in variations in diameter (see above), with some sections being notably wider or narrower than others. To mitigate this issue, we calculated the average of the longest and narrowest segments for both the myelinated and raw axon diameters. These values were also used to compute the g-ratio.
It is possible that the irregular morphology of these axons contributes to the unexpected findings in terms of axon diameter and myelin thickness, and consequently, the g-ratio. While a reduction in the g-ratio typically indicates an increase in myelin thickness, the unique structure of these axons may limit the potential for increased myelination while still resulting in a lower g-ratio due to their inconsistent shape.
Further exploration is warranted, potentially through multiple cross-sections or 3D modelling of these axons. A clearer understanding of their morphology could enhance our insights into their myelination and the interplay of these factors. We have highlighted this in discussion.
Perform the TEM experiment in P90 mice, as other experiments were conducted in both P21 and P90 mice.
Response: We sincerely appreciate the reviewer’s insightful comments. Immunofluorescence staining and transmission electron microscopy (TEM) necessitate distinct postfixing methods, specifically using either paraformaldehyde or a combination of paraformaldehyde and glutaraldehyde. Unfortunately, we faced a shortage of mice, which prevented us from conducting the full experiment. Consequently, we opted to perform immunofluorescence postfixing on the brains of the limited number of P90 mice available. This approach enabled us to gain valuable insights into the biological mechanisms that regulate developmental myelination.
By adopting this approach, we were able to conduct a wider array of analyses, including oligodendrocyte quantification, myelination assessment, and evaluation of astrocyte presence. Given that the immunofluorescence data at P21 correlates with the TEM data, we expect to observe a similar trend at P90. Nevertheless, we recognize that performing TEM at P90 would be a valuable addition. That said, we believe it may not substantially enhance the manuscript's findings and could potentially delay publication due to the extended timelines for breeding and analysis required. We have acknowledged this limitation in our discussion and remain receptive to the reviewer's perspective on its necessity for the paper, though additional time would be required.
Upon reviewing the overall results, I noticed that in 16p11.2 deletion mice, the number of Ols was unchanged (Figure 2), while myelin thickness increased (Figure 4e). In contrast, in 16p11.2 duplication mice, the number of OLs increased (Figure 2), but myelin thickness remained unchanged (Figure 4e). These are curious findings.
Response: Thank you for bringing up this important point. The observation of no increase in the number of oligodendrocytes, coupled with a thicker myelin sheath around myelinated axons, may be attributed to the efficiency or strength of the interaction between axons and oligodendrocytes. This relationship is crucial for effective myelination and warrants further molecular exploration in future research, particularly to assess the effect of neuronal maturity and its associated effect on myelination across experimental groups.
Interestingly, while the myelin in deletion mice is thicker, the percentage of myelinated axons (Figure 4b) decreases. This could indicate that the number of axons contacted by each oligodendrocyte is reduced, although the connections formed may be stronger, leading to increased myelination of targeted axons.
Conversely, in duplication mice, despite an apparent increase in oligodendrocytes and a decrease in myelin thickness, the percentage of myelinated axons is significantly higher (Figure 4b). This suggests that the elevated number of oligodendrocytes, as shown in Figure 2, results in a greater percentage of axons being myelinated (Figure 4b). However, the observed reduction in myelin thickness might be attributed to a diminished capacity of these oligodendrocytes to effectively myelinate irregularly formed axons, which could explain the intriguing results we present.
Astrocyte data should be moved to the supplemental material since the main text focused on OLs and myelination changes in 16p11.2 mice. Additionally, ensure the asterisk is displayed in Figure 5.
Response: Thank you for your feedback. We have now added the appropriate asterisks indicating the level of significance to Figure 5. Given the potential influence of astrocytes on myelination capacity (PMID: 26988764; 37291151), we believe this figure should remain in the main text. It is essential to emphasize the role of this critical cell type in myelination, even if indirectly. Furthermore, considering the previous comments and our responses, we believe that having this in the main text may enhance the paper's overall readability. We have ensured that asterisk is clearly visible.
Capitalise protein names throughout the manuscript.
Response: Thank you for bringing this to our attention. All protein names have now been capitalized.
We sincerely appreciate the time you took to read our manuscript and provide valuable feedback that has enhanced its quality. We have thoughtfully addressed each of your comments (noted in blue) and revised the manuscript accordingly. Below, you will find our responses (in black) to your suggestions.

Reviewer 2 Report
Comments and Suggestions for Authors
The authors present an interesting work on the effects of a fairly frequent form of genomic copy number variation on biological processes associated with white matter.
The CNVs studied are those involving the 16p11.2 chromosome region and the authors use mice with genomic imbalances corresponding to the human 16p11.2 microdeletion and microduplication as a model.
The study appears well-designed and correctly conducted. The results obtained are interesting and are presented clearly, despite their complexity. The only limitations, also correctly pointed out by the authors, concern the small number of subjects examined and that the study was conducted on a mouse model, although validated, so the results need to be confirmed in humans.
I just have a few specific comments to make.
On page 2, lines 85-88: the sentence “Myelin is produced by mature oligodendrocytes, a complex process involving specification of neural progenitor cells into oligodendrocyte progenitor cells which then proliferate, migrate, and differentiate into mature, myelin producing oligodendrocytes.” seems to have some problems.
On page 6, line 213: "(t = 14.21, p < 00.1)” should be "(t = 14.21, p < 0.001)".
On page 7, line 239: “t = 14.21, p < 00.1)” should be “t = 14.21, p < 0.001)”.
On page 7, figure 5: the symbols for the significance level of the observed differences are missing.
On page 8, line 282 "versus the mice." should be "versus the wild type mice."
On page 10, lines 412-413: “are commonly utilized and to understand” should be “are commonly utilized to understand”.
On page 13, line 536: revise “GraphPad Prism (version 10.1.2 (324)).”.
Pages 13-15: the whole paragraph 7 seems to be a duplicate of paragraph 6.
Author Response
Reviewer 2
The authors present an interesting work on the effects of a fairly frequent form of genomic copy number variation on biological processes associated with white matter.
The CNVs studied are those involving the 16p11.2 chromosome region and the authors use mice with genomic imbalances corresponding to the human 16p11.2 microdeletion and microduplication as a model.
The study appears well-designed and correctly conducted. The results obtained are interesting and are presented clearly, despite their complexity. The only limitations, also correctly pointed out by the authors, concern the small number of subjects examined and that the study was conducted on a mouse model, although validated, so the results need to be confirmed in humans.
I just have a few specific comments to make.
We sincerely appreciate the reviewer’s time and effort in evaluating our manuscript and providing positive feedback. We are delighted to learn that the reviewer found our paper engaging, well-structured, and clear, with thorough coverage of its limitations. We also thank the reviewer for their specific comments, which we have addressed accordingly and highlighted in red within the manuscript document.
On page 2, lines 85-88: the sentence “Myelin is produced by mature oligodendrocytes, a complex process involving specification of neural progenitor cells into oligodendrocyte progenitor cells which then proliferate, migrate, and differentiate into mature, myelin producing oligodendrocytes.” seems to have some problems.
Response: We appreciate the reviewer for highlighting an issue with this sentence. It has now been revised to: “Myelin is produced by mature oligodendrocytes through a complex process that begins with the specification of neural progenitor cells into oligodendrocyte progenitor cells. These progenitor cells then proliferate, migrate, and differentiate into mature oligodendrocytes, which ultimately produce myelin.” We hope this clarification addresses the concerns raised.
On page 6, line 213: “(t = 14.21, p < 00.1)” should be “(t = 14.21, p < 0.001)”.
Response: Thank you for noticing this, it has been rectified.
On page 7, line 239: “(t = 14.21, p < 00.1)” should be (“t = 14.21, p < 0.001)”.
Response: This has also been rectified.
On page 7, figure 5: the symbols for the significance level of the observed differences are missing.
Response: Thank you for noticing this. The correct significance level symbols have now been added to Figure 5.
On page 8, line 282 “versus the mice.” should be “versus the wild-type mice.”
Response: Thank you for pointing this out. The phrase has now been rectified.
On page 10, lines 412-413: “are commonly utilized and to understand” should be “are commonly utilized to understand”.
Response: Thank you for identifying this, it has now been rectified.
On page 13, line 536: revise “GraphPad Prism (version 10.1.2 (324)).”.
Response: This has now been updated to 10.2.3 which is correct.
Pages 13-15: the whole paragraph 7 seems to be a duplicate of paragraph 6.
Response: Yes, thank you for highlighting this. The entirety of paragraph 7 (from line 537 – 628) has now been removed.

Round 2
Reviewer 1 Report
Comments and Suggestions for Authors
1. The statistical analysis is problematic, as the authors did not specify the statistical methods used. An example of appropriate statistical reporting would be:
"All data are expressed as the Mean ± Standard Error of the Mean (SEM). Statistical analyses were conducted using Prism (V9, GraphPad Software, USA). Data distribution was assessed initially using the Shapiro-Wilk test to determine suitability for parametric or nonparametric tests. An unpaired t-test was employed for normally distributed data in two-group comparisons. Non-normally distributed data were analyzed using the Mann-Whitney U test. The paired-pulse ratio was evaluated using two-way repeated measures ANOVA with Sidak's post hoc tests. Specific experiment details, including exact sample sizes (n), precision measures, statistical tests performed, and definitions of significance, are provided in figure legends. Statistical significance was set at P < 0.05."
2. The OLIG2 staining data are unreliable, as the authors did not use a new OLIG2 antibody for the immunostaining experiments.
3. The transmission electron microscopy (TEM) data are also unreliable. In Figure 4e, the authors state that myelin thickness is reduced in 16p11.2 duplication mice, but the mean values for wild-type and duplication mice appear to be the same. Additionally, in Figure 4f, the g-ratio is decreased in duplication mice, which indicates increased myelin thickness. These conflicting results suggest the TEM analysis was problematic.
4. The authors' excuse for not performing TEM at P90 due to a lack of 16p11.2 mice is not a reasonable justification, as the authors should have planned their experiments accordingly to ensure sufficient sample sizes.
5. The asterisks indicating the level of significance are still not labeled in Figure 5.
In summary, the statistical analysis, OLIG2 staining, TEM data, and figure labeling all have significant issues that need to be addressed by the authors. The concerns raised in the original feedback have not been adequately resolved.
Author Response
- The statistical analysis is problematic, as the authors did not specify the statistical methods used. An example of appropriate statistical reporting would be:"All data are expressed as the Mean ± Standard Error of the Mean (SEM). Statistical analyses were conducted using Prism (V9, GraphPad Software, USA). Data distribution was assessed initially using the Shapiro-Wilk test to determine suitability for parametric or nonparametric tests. An unpaired t-test was employed for normally distributed data in two-group comparisons. Non-normally distributed data were analyzed using the Mann-Whitney U test. The paired-pulse ratio was evaluated using two-way repeated measures ANOVA with Sidak's post hoc tests. Specific experiment details, including exact sample sizes (n), precision measures, statistical tests performed, and definitions of significance, are provided in figure legends. Statistical significance was set at P < 0.05." Response: Thank you for bringing this to our attention. We have updated the legend as per the reviewer's suggestion.
- The OLIG2 staining data are unreliable, as the authors did not use a new OLIG2 antibody for the immunostaining experiments. Response: We have completed the staining with anti-OLIG2 (AB109186, Abcam) as recommended by the reviewer. The updated images and analysis are now presented in Figure 2, highlighting the nuclear Olig2 staining.
- The transmission electron microscopy (TEM) data are also unreliable. In Figure 4e, the authors state that myelin thickness is reduced in 16p11.2 duplication mice, but the mean values for wild-type and duplication mice appear to be the same. Additionally, in Figure 4f, the g-ratio is decreased in duplication mice, which indicates increased myelin thickness. These conflicting results suggest the TEM analysis was problematic. Response: We acknowledge the reviewer’s observation regarding the confusing results. To address this, we reanalyzed the data with a blinded evaluator. The updated analysis indicates an increase in both the percentage of myelinated axons and myelin thickness in 16p11.2 duplication mice at P30. These findings are now presented in Figure 5.
- The authors' excuse for not performing TEM at P90 due to a lack of 16p11.2 mice is not a reasonable justification, as the authors should have planned their experiments accordingly to ensure sufficient sample sizes. Response: In response to the reviewer’s suggestion, we attempted to breed the 16p11.2 duplication mice. However, we encountered significant technical and logistical challenges during the breeding process. Although we made efforts to proceed, the genotyping results were inconclusive, possibly due to accidental mix-ups of the mice. Currently, we are in the process of acquiring new colonies, which will take approximately a year to establish. As a result, we are unable to perform TEM analysis on P90 mice at this time. Additionally, since our findings already demonstrate a downregulation of MBP expression in P90 mice (Figure 3), conducting TEM would primarily validate this result without contributing new insights to the study. For these reasons, we sincerely apologize for being unable to fulfill this request. We appreciate your understanding of these limitations. We have also acknowledged this limitation in discussion.
- The asterisks indicating the level of significance are still not labeled in Figure 5. Response: Thanks for pointing this out and apologies for missing this out. We have now added it to the figure 5.
In summary, the statistical analysis, OLIG2 staining, TEM data, and figure labeling all have significant issues that need to be addressed by the authors. The concerns raised in the original feedback have not been adequately resolved.
Response: We sincerely thank the reviewer for their time and constructive feedback, which has significantly improved the quality of our manuscript. We have made every effort to address all comments to the best of our ability, except for the TEM data for d90. Despite our best efforts, technical challenges have prevented us from generating new mice, and realistically, it would not be feasible to obtain this data within a year. As this data is complementary and would not provide additional insights to the study, we kindly request the reviewer to consider our manuscript favorably.
Round 3
Reviewer 1 Report
Comments and Suggestions for Authors
1 In the part of the TEM data, there are several errors:
a It seems that unpaired t test was used instead of One-way ANOVA, because t value was shown along with the p value.
b In the figure 4 legend, the sequence number should be f, g.
c in the figure 4 results, the sequecence number is very confusing.
d in figure 4e, the g-ratio was both reduced in the two groups, which means reduced myelin thickness, but the description was not correct in the result part.
e representative image of enlarged myelin should be shown in figure 4.
2 In the whole manuscript, unpaired t test was used instead of One-way ANOVA, because t value was shown along with the p value.
3 in the figure 5, the p value was not labeled.
Author Response
We sincerely thank the Editor and reviewer(s) for their valuable time and constructive feedback. All comments have been carefully addressed, and we hope you find the revised version suitable for publication in IJMS.
point-by-point response:
1 In the part of the TEM data, there are several errors:
a It seems that unpaired t test was used instead of One-way ANOVA, because t value was shown along with the p value. Response: Thanks for bringing this to our attention and agree that two values could be confusing. As per suggestion, so we re-analysed the data by One-way ANOVA. We have updated the p values.
b In the figure 4 legend, the sequence number should be f, g. Response: Thank you, the order has been changed.
c in the figure 4 results, the sequecence number is very confusing. Response: We appreciate the reviewer’s feedback about the confusing results. The revised sections have now been clarified and updated accordingly.
d in figure 4e, the g-ratio was both reduced in the two groups, which means reduced myelin thickness, but the description was not correct in the result part. Response: Thank you for bringing this to our attention. We have revised this section, and it is now included in the results portion (lines 223-235).
e representative image of enlarged myelin should be shown in figure 4. Response: Based on your suggestion, we have updated the images, which are now presented in Figure 4a.
2 In the whole manuscript, unpaired t test was used instead of One-way ANOVA, because t value was shown along with the p value. Response: Thank you for bringing this to our attention. Following your suggestion, we have thoroughly revised the manuscript and included only p-values.
3 in the figure 5, the p value was not labelled. Response: Thank you for bringing this to our attention, and apologies for the oversight. We have now updated Figure 5 to include it.
Round 4
Reviewer 1 Report
Comments and Suggestions for Authors
All the concerns are solved.